# Occupational bladder cancer: A cross section survey of previous employments, tasks and exposures matched to cancer phenotypes

Oliver Reed[1], Ibrahim Jubber[1], Jon Griffin[2], Aidan P. Noon[3], Louise Goodwin[1], Syed Hussain[4], Marcus G. Cumberbatch[1,5‡], James W.F. Catto[1,3‡]*

**1** Academic Urology Unit, University of Sheffield, Sheffield, United Kingdom, **2** Department of Pathology, Sheffield Teaching Hospitals NHS Trust, Sheffield, United Kingdom, **3** Department of Urology, Sheffield Teaching Hospitals NHS Trust, Sheffield, United Kingdom, **4** Academic Oncology Unit, Western Park, University of Sheffield, Sheffield, United Kingdom, **5** Peter MacCallum Cancer Centre, Melbourne, Australia

‡ These authors share last authorship on this work.
* j.catto@sheffield.ac.uk

## Abstract

### Objectives

Up to 10% of Bladder Cancers may arise following occupational exposure to carcinogens. We hypothesised that different cancer phenotypes reflected different patterns of occupational exposure.

### Methods

Consecutive participants, with bladder cancer, self-completed a structured questionnaire detailing employment, tasks, exposures, smoking, lifestyle and family history. Our primary outcome was association between cancer phenotype and occupational details.

### Results

We collected questionnaires from 536 patients, of whom 454 (85%) participants (352 men and 102 women) were included. Women were less likely to be smokers (68% vs. 81% Chi sq. p<0.001), but more likely than men to inhale environmental tobacco smoke at home (82% vs. 74% p = 0.08) and use hair dye (56% vs. 3%, p<0.001). Contact with potential carcinogens occurred in 282 (62%) participants (mean 3.1 per worker (range 0–14)). High-grade cancer was more common than low-grade disease in workers from the steel, foundry, metal, engineering and transport industries (p<0.05), and in workers exposed to crack detection dyes, chromium, coal/oil/gas by-products, diesel fumes/fuel/aircraft fuel and solvents (such as tri-chloroethylene). Higher staged cancers were frequent in workers exposed to Chromium, coal products and diesel exhaust fumes/fuel (p<0.05). Various workers (e.g. exposed to diesel fuels or fumes (Cox, HR 1.97 (95% CI 1.31–2.98) p = 0.001), employed in a garage (HR 2.19 (95% CI 1.31–3.63) p = 0.001), undertaking plumbing/gas fitting/ventilation (HR 2.15 (95% CI 1.15–4.01) p = 0.017), undertaking welding (HR 1.85 (95% CI 1.24–2.77) p = 0.003) and exposed to welding materials (HR 1.92 (95% CI 1.27–2.91) p = 0.002)) were more likely to

**Data Availability Statement:** We are unable to release the full data set because it contains information that would allow individuals to be identified. Specifically, the granularity of detail is such that persons could cross gender, age,

regional residence and occupational combinations to identify specific individuals with bladder cancer and their family history. Data access queries may be directed to Sarah Bottomley (contact via s.e. bottomley@sheffield.ac.uk).

**Funding:** JWFC received project grant awards to perform this work from Yorkshire Cancer Research (yorkshirecancerresearch.org.uk, Numbers S310: Epigenetic carcinogenesis in the urothelium: development of a model system and examination of candidate occupational carcinogens and S385: The Yorkshire Cancer Research Bladder Cancer Patient Reported Outcomes Survey). The funders had no role in design and conduct of the study; collection, management, analysis, and interpretation of the data; preparation, review, or approval of the manuscript; and decision to submit the manuscript for publication.

**Competing interests:** The authors have read the journal's policy and have the following potential competing interests: SH has received research funding from CR UK, MRC/NIHR, UHB charities, CCC charities, North West Cancer Research and reimbursement for consultancy from Bayer, Janssen, Boehringer Ingelheim, Pierre Fabre, Eli Lilly. SAH Advisory board/Consultancy: Roche, MSD, AstraZeneca, BMS, Janssen, Pfizer, Astellas, Bayer, Pierre Fabre, Sotio, GSK, Ipsen and Eisai. JWFC has received reimbursement for consultancy from Astra Zeneca, Roche and Janssen, speaker fees from BMS, MSD, Nucleix and Roche, and honoraria for membership of an advisory board for Ferring. This does not alter our adherence to PLOS ONE policies on sharing data and materials. There are no patents, products in development or marketed products associated with this research to declare.

have disease progression and receive radical treatment than others. Fewer than expected deaths were seen in healthcare workers (HR 0.17 (95% CI 0.04–0.70) p = 0.014).

## Conclusions

We identified multiple occupational tasks and contacts associated with bladder cancer. There were some associations with phenotype, although our study design precludes robust assessment.

## Introduction

Bladder cancer (BC) is a common human malignancy and one of the most expensive to manage [1]. Most tumours present with haematuria [2] and at diagnosis around 30% are muscle invasive and 70% non-muscle invasive cancers (NMI) [3]. NMI tumours are stratified into low and high grade lesions, to reflect different treatments and outcomes [4]. The majority of BCs are urothelial cell carcinoma (UCC) in histological sub-type and arise following exposure to carcinogens excreted in the urine [5]. The most common bladder carcinogens are found through tobacco smoke [6] or occupation task [7,8]. Risk from smoking varies with gender, duration, tobacco type and mode of inhalation [9,10]. These aetiological factors mean that BCs are most common in older patients, in men and in the Western World [1]. An individual's risk of BC reflects their carcinogen burden and their ability to metabolise pro-carcinogens [11].

Around 10% of BCs arise following occupational exposure to carcinogens [12]. These carcinogens may be broadly classified into *aromatic amines*, *polycyclic aromatic hydrocarbons* (PAHs), *heavy metals* or *mixed compounds* [7]. The occupational exposure of workers to many carcinogens has been limited by health and safety regulations [such as the European Union directives (e.g. 90/394/EEC and 98/24/EC) and the 2002 Control of Substances Hazardous to Health Regulations in the UK] and changes in manufacturing. Whilst many high risk urothelial carcinogens have been identified, it is suspected that more are still in use. The uncertainty about and identification of further candidates reflects the long latency between exposure and cancer, variations in an individual's risk, that many workers also smoke, and that many potential carcinogens are in widespread (such as diesel fumes) or occult use [13].

BC arises in at least two distinct phenotypes, reflecting genomic events [14,15]. Low-grade tumours are characterised by papillary growth patterns, few genetic alterations (e.g. FGFR3 or hTERT mutation) and an indolent behaviour [16]. In contrast, high-grade BC is an aggressive disease with genetic and epigenetic instability [17], and multiple mutations [18]. We hypothesised that the BC phenotypes could reflect different carcinogenic exposures and, in turn, occupational tasks. We explored this hypothesis using a large Scandinavian dataset and found various occupations with different risks for localised and invasive BCs, and higher rates of BC mortality in the building sector [19]. However, this dataset lacked granularity of occupational tasks, personal smoking exposure and classified BC by stage not grade of differentiation.

To build upon our prior work, we undertook a prospective detailed occupation survey using a consecutive cohort of patients arising in a region of high BC risk. We annotated patients with detailed histological and outcome data.

## Materials and methods

### Patients and occupational questionnaire

Consecutive patients with a new diagnosis of BC treated at the Royal Hallamshire Hospital, Sheffield (RHH), were enrolled from February 2010 to July 2012. RHH is the sole urological

service in Sheffield (population 600,000) and the cancer center for South Yorkshire, UK (population 1.9 million). Participants self-completed a structured questionnaire containing questions on employment history, occupational tasks (nature and frequency) and exposures [S1 Fig] over their whole lifetime. The questionnaire was designed in collaboration with Sheffield Occupational health Advisory Service (SOHAS) after systematic review [7,8] and included sections for smoking (direct and passive environmental tobacco smoke (ETS)) [9] hobbies linked to BC [20,21], lifestyle and family cancer history. Patients with non-urothelial BC (e.g. squamous cell or adenocarcinoma) were excluded due to different causative associations. Paper questionnaires were completed at home and returned using a stamp addressed envelope, before uploading to a prospective database. All patients gave informed consent in an ethically approved programme (South Yorkshire Research Ethics Committee approval number 10/H1310/73) agreed by Sheffield Teaching Hospital review board. Occupational classes were assigned using NYK and ISCO-1958 codes (as detailed in [7]). In persons with multiple occupations or those with short duration we selected the 3 occupations of longest duration and a minimum period of 1 year as previously validated [19]. No formal power calculation was performed. This study was an explorative cohort study and so we included all eligible patients in the recruiting time frame and aimed to describe data (to allow future studies to be powered accordingly).

## Pathological and clinical outcomes

Tumours were classified by specialist uropathologists using the 1973 WHO and TNM criteria [22]. In participants with multiple BCs, we analysed outcomes with respect to the primary BC. Patients were treated according to local network (http://www.northtrentcancernetwork.nhs.uk/urology.htm) and international guidelines [3]. Outcome data were collected between August and October 2018 using hospital databases [namely Integrated Clinical Environment (ICE), Lorenzo and EDMS software]. We measured tumour behaviour with respect to time following initial treatment and defined recurrence as a subsequent NMI cancer following a similar tumour and progression as an increase in pathological stage. Radical treatment was measured to the date of Radical Surgery or starting Radical radiotherapy. Date of death was defined using death certification.

## Statistical analysis

Our primary outcome was the association between BC phenotype (measured as Grade and Stage) and occupational sector, task and exposures. Secondary outcomes were occupational associations with local recurrence, disease progression, radical treatment and mortality. Data were analysed according to participant self-reported questionnaires. Data cleaning clarified missing or unclear parameters, but did not alter returns. Comparisons between occupational exposures and patient/tumour features were performed using Chi-squared tests for categorical and Students T or Mann Whitney U tests for continuous data. Correlation was determined using Pearson's coefficient. Survival was plotted against time using the Kaplan-Meier method and compared using Cox regression analysis. Patients were censored at last follow up or death. All analysis was performed in SPSS software (version 24.0, SPSS Corp). Statistical tests were two-tailed and significance defined as $p < 0.05$.

## Patient and public involvement

The idea for this project arose following discussions with Simon Pickvance, Sheffield Occupational Health Advisory Service (SOHAS) and local patients. SOHAS works with employees affected by occupational health problems and with employers to improve occupational

hygiene. We had observed patients with BC and unusual employment tasks (such as the use of crack detection dyes [13]) or high levels of exposure to heavy metals (in soldering or welding tasks). The occupational questionnaire was designed with SOHAS and refined over several iterations using small patient groups.

## Results

### Patients and tumours

We collected questionnaires from 536 patients, of whom 82 (15%) were excluded as they had either non-urothelial BCs, non-primary BCs, missing follow up (e.g. in another hospital) or histopathological details were incomplete. We had sufficient data on 454 (85%) participants (Table 1), including 352 (78%) men and 102 women (22%). In total, 25% had a first degree relative with cancer and 355 (78%) participants were smokers (including 118 smoking at BC diagnosis). Women were less likely to be smokers (68% vs. 81% Chi sq. p<0.001), but more likely to inhale ETS at home (82% vs. 74% p = 0.08) and use hair dye (56%, p<0.001) than men. Hobbies varied considerably between the sexes, with more men undertaking fishing and model building (Chi sq. p<0.001). BCs were distributed evenly between low (140 (31%)), moderate (140 (31%)) and high grade (174, 38%) lesions. With regards to stage, most cancers were NMI (368 (88%)) at diagnosis, including 191 (42%) that were high risk (either pTis, pT1 or Grade 3 pTa). Following treatment, recurrence was seen in 244/447 (56%), progression in 114/448 (25%), radical treatment in 156/450 (35%) and death in 157/451 (35%) participants at median of 101 months (interquartile range 73–128).

### Employers and occupational class

Individual employers were documented in 393 (87%) participants, including an average of 3.2 (St dev. ± 2.7) each for men and 2.3 (± 2.1) for women (T Test p = 0.003). There were considerable differences in employment class between men and women (Table 2). The most common male occupations were in engineering, steel and metal working sectors (40%). Women most commonly worked in the service industries (25%). High grade BC was more common than low grade BC in workers from the steel, foundry, metal, engineering and transport industries (Table 2, p<0.05). With regards to stage, engineering and metal workers had higher than expected risks of high-risk NMIs BCs (pTis and pT1, chi sq. p = 0.02).

### Substance exposures

Contact with potential or confirmed bladder carcinogens was reported by 282 (62%) participants (mean 3.1 per worker (range 0–14), Table 3). There were marked differences between the genders reflecting employment patterns. The most common contacts were diesel fumes/fuel (n = 176 (39%)), coal/oil/gas by-products (151 (33%)), solvents (125 (28%)), heavy metals (50 (11%)), coking plant fumes (40 (9%)) and crack detection dyes (31 (7%)). Relatively few participants were exposed to more typical urothelial carcinogens such as textile dyes (28 (6%)), printing inks (30 (7%)), 4-aminobiphenyl/MOCA/DDM/MDA/o-toluidine (4 (1%)) reflecting the manufacturing sectors in Yorkshire. Participants often had contact with multiple, similar substances (e.g. diesel fumes (21%) and diesel fuel (18%, Pearson's correlation r = 0.80, p<0.001), Cadmium (4%) and Chromium (8%, Pearson's correlation r = 0.47, p<0.001)). High grade BC was more common than low grade cancer in workers exposed to crack detection dyes, chromium, coal/oil/gas by-products, diesel fumes/fuel/aircraft fuel and solvents (such as trichloroethylene). Higher staged cancers were more frequent than expected in workers exposed to Chromium, coal products and diesel exhaust fumes/fuel (p≦0.05).

**Table 1. Patients and tumours in this report.**

| | | Male | | Female | | Total | | Chi Sq. P |
|---|---|---|---|---|---|---|---|---|
| Age at diagnosis (Mean ± St dev) | | 67.0 ±9.3 | | 66.4 ± 11.3 | | | | T Test p = 0.58 |
| First degree relative with cancer | 0 | 264 | 75% | 78 | 77% | 342 | 75% | |
| | 1 | 64 | 18% | 15 | 15% | 79 | 17% | |
| | 2 | 19 | 5% | 7 | 7% | 26 | 6% | |
| | 3 or more | 5 | 1% | 2 | 2% | 7 | 2% | 0.843 |
| Smoking history | Non-smoker | 66 | 19% | 33 | 32% | 99 | 22% | |
| | Smoker | 286 | 81% | 69 | 68% | 355 | 78% | **0.003** |
| Years smoking (Mean ± St dev) | | 36.7 ± 17.8 | | 34.7 ± 18.2 | | | | T Test p = 0.39 |
| Pack years (Mean ± St dev) | | 35.3 ± 27.4 | | 25.8 ± 20.0 | | | | T Test p = **0.008** |
| ETS at home | Yes | 260 | 74% | 84 | 82% | 344 | 76% | |
| | No | 92 | 26% | 18 | 18% | 110 | 24% | 0.078 |
| ETS at work | Yes | 281 | 80% | 70 | 69% | 351 | 77% | |
| | No | 71 | 20% | 32 | 31% | 103 | 23% | **0.017** |
| Fishing | Yes | 100 | 28% | 7 | 7% | 107 | 24% | |
| | No | 252 | 72% | 95 | 93% | 347 | 76% | <**0.001** |
| Swimming | Yes | 104 | 30% | 36 | 35% | 140 | 31% | |
| | No | 248 | 71% | 66 | 65% | 314 | 69% | 0.270 |
| Model building | Yes | 53 | 15% | 1 | 1% | 54 | 12% | |
| | No | 299 | 85% | 101 | 99% | 400 | 88% | <**0.001** |
| Hair dye use | Yes | 10 | 3% | 57 | 56% | 67 | 15% | |
| | No | 342 | 97% | 45 | 44% | 387 | 85% | <**0.001** |
| Phenacetin | Yes | 9 | 3% | 3 | 3% | 12 | 3% | |
| | No | 343 | 97% | 99 | 97% | 442 | 97% | 0.830 |
| Coal tar creams | Yes | 16 | 5% | 3 | 3% | 19 | 4% | |
| | No | 336 | 96% | 99 | 97% | 435 | 96% | 0.480 |
| UCC Grade | 1 | 88 | 25% | 52 | 51% | 140 | 31% | |
| | 2 | 115 | 33% | 25 | 25% | 140 | 31% | |
| | 3 | 149 | 42% | 25 | 25% | 174 | 38% | **0.020** |
| Presence of CIS | Yes | 46 | 13% | 4 | 4% | 50 | 11% | |
| | No | 305 | 87% | 98 | 96% | 403 | 89% | <**0.001** |
| UCC Stage | pTa | 216 | 62% | 77 | 76% | 293 | 65% | |
| | pTis | 15 | 4% | 0 | 0% | 15 | 3% | |
| | pT1 | 75 | 21% | 13 | 13% | 88 | 20% | |
| | pT2-4 | 44 | 13% | 12 | 12% | 56 | 12% | **0.020** |
| | Total | 352 | 78% | 102 | 22% | 454 | 100% | |

## Occupational task

We questioned participants about their direct involvement or close proximity to ('nearby') 33 tasks thought to potentially reflect exposure to urothelial carcinogens (Table 4). Tasks were selected from SOHAS prior occupational cancer episodes. In total, 1,370 tasks were identified by 210 participants. The commonest tasks were welding (n = 115 (25%)), making cement (94 (21%), using lubricating/coolant oils (97 (21%)), soldering/brazing (93 (20%)), degreasing (90 (20%)) or involved inhaling fumes from quenching/forging or cooling (174 (38%)). As with substance exposures and occupational class, there were differences in tasks between the sexes and the associated BCs. Cancers of higher than expected grades were seen with welding, the use of mineral oil lubricants, the use of protective resins and with tasks that included diesel

**Table 2. Occupational class compared to patient sex and tumour phenotype.**

| | Gender | | | Grade | | | | Stage | | | | |
|---|---|---|---|---|---|---|---|---|---|---|---|---|
| | Male | Female | Chi sq P | 1 | 2 | 3 | Chi sq P | Ta | Tis | T1 | T2-4 | Chi sq P |
| Coke, coal, power generation | 60 (100%) | 0 (0%) | <0.001 | 13 (21.6%) | 21 (35%) | 26 (43.3%) | 0.25 | 34 (57.6%) | 2 (3.3%) | 13 (22%) | 10 (16.9%) | 0.58 |
| Coking plant or gas production | 34 (100%) | 0 (0%) | | 6 (17.6%) | 12 (35.2%) | 16 (47%) | | 21 (63.6%) | 2 (6%) | 6 (18.1%) | 4 (12.1%) | |
| Coal mining/smokeless fuel making | 37 (100%) | 0 (0%) | | 9 (24.3%) | 11 (29.7%) | 17 (45.9%) | | 20 (54%) | 1 (2.7%) | 9 (24.3%) | 7 (18.9%) | |
| Nuclear power | 7 (100%) | 0 (0%) | | 2 (28.5%) | 1 (14.2%) | 4 (57.1%) | | 4 (57.1%) | 0 (0%) | 1 (14.2%) | 2 (28.5%) | |
| Steel and foundry | 139 (92.6%) | 11 (7.3%) | <0.001 | 36 (23.6%) | 54 (35.5%) | 62 (40.7%) | 0.05 | 90 (59.6%) | 8 (5.2%) | 35 (23.1%) | 18 (11.9%) | 0.15 |
| Metal refining | 26 (100%) | 0 (0%) | | 4 (15.3%) | 9 (34.6%) | 13 (50%) | | 15 (57.6%) | 2 (7.6%) | 6 (23%) | 3 (11.5%) | |
| Steel industry | 100 (93.4%) | 7 (6.5%) | | 27 (24.7%) | 35 (32.1%) | 47 (43.1%) | | 59 (54.6%) | 7 (6.4%) | 27 (25%) | 15 (13.8%) | |
| Steel production | 62 (93.9%) | 4 (6%) | | 18 (27.2%) | 18 (27.2%) | 30 (45.4%) | | 41 (62.1%) | 4 (6%) | 12 (18.1%) | 9 (13.6%) | |
| Heat treatment | 50 (94.3%) | 3 (5.6%) | | 15 (28.3%) | 13 (24.5%) | 25 (47.1%) | | 33 (62.2%) | 4 (7.5%) | 11 (20.7%) | 5 (9.4%) | |
| Forging | 39 (95.1%) | 2 (4.8%) | | 12 (29.2%) | 11 (26.8%) | 18 (43.9%) | | 24 (58.5%) | 3 (7.3%) | 9 (21.9%) | 5 (12.1%) | |
| Foundries | 53 (98.1%) | 1 (1.8%) | | 12 (21.8%) | 19 (34.5%) | 24 (43.6%) | | 31 (56.3%) | 3 (5.4%) | 14 (25.4%) | 7 (12.7%) | |
| Engineering and metals | 140 (90.3%) | 15 (9.6%) | <0.001 | 37 (23.4%) | 51 (32.2%) | 70 (44.3%) | 0.03 | 92 (58.5%) | 9 (5.7%) | 39 (24.8%) | 17 (10.8%) | 0.02 |
| Engineering | 84 (91.3%) | 8 (8.6%) | | 22 (23.1%) | 36 (37.8%) | 37 (38.9%) | | 55 (58.5%) | 5 (5.3%) | 21 (22.3%) | 13 (13.8%) | |
| Electroplating | 8 (88.8%) | 1 (11.1%) | | 2 (22.2%) | 1 (11.1%) | 6 (66.6%) | | 6 (66.6%) | 1 (11.1%) | 2 (22.2%) | 0 (0%) | |
| Cutlery | 25 (75.7%) | 8 (24.2%) | | 9 (27.2%) | 7 (21.2%) | 17 (51.5%) | | 22 (66.6%) | 0 (0%) | 9 (27.2%) | 2 (6%) | |
| Welding | 54 (100%) | 0 (0%) | | 5 (8.9%) | 23 (41%) | 28 (50%) | | 28 (50%) | 6 (10.7%) | 17 (30.3%) | 5 (8.9%) | |
| Making electrical contacts/solder | 27 (93.1%) | 2 (6.8%) | | 11 (37.9%) | 8 (27.5%) | 10 (34.4%) | | 22 (75.8%) | 1 (3.4%) | 3 (10.3%) | 3 (10.3%) | |
| Soldering | 48 (92.3%) | 4 (7.6%) | | 16 (30.1%) | 18 (33.9%) | 19 (35.8%) | | 31 (58.4%) | 2 (3.7%) | 13 (24.5%) | 7 (13.2%) | |
| Other manufacturing | 70 (89.7%) | 8 (10.2%) | 0.01 | 21 (26.9%) | 19 (24.3%) | 38 (48.7%) | 0.11 | 45 (57.6%) | 4 (5.1%) | 18 (23%) | 11 (14.1%) | 0.46 |
| Refining and recycling | 7 (100%) | 0 (0%) | | 2 (28.5%) | 0 (0%) | 5 (71.4%) | | 5 (71.4%) | 1 (14.2%) | 0 (0%) | 1 (14.2%) | |
| Making garments & textiles | 3 (33.3%) | 6 (66.6%) | | 4 (44.4%) | 1 (11.1%) | 4 (44.4%) | | 5 (55.5%) | 0 (0%) | 3 (33.3%) | 1 (11.1%) | |
| Spinning synthetic fibre | 1 (50%) | 1 (50%) | | 1 (50%) | 0 (0%) | 1 (50%) | | 1 (50%) | 0 (0%) | 1 (50%) | 0 (0%) | |
| Plastic production | 17 (100%) | 0 (0%) | | 4 (23.5%) | 5 (29.4%) | 8 (47%) | | 10 (58.8%) | 1 (5.8%) | 4 (23.5%) | 2 (11.7%) | |
| Cement products | 30 (100%) | 0 (0%) | | 7 (23.3%) | 9 (30%) | 14 (46.6%) | | 17 (56.6%) | 1 (3.3%) | 9 (30%) | 3 (10%) | |
| Rubber tyre industorry | 7 (100%) | 0 (0%) | | 1 (14.2%) | 2 (28.5%) | 4 (57.1%) | | 6 (85.7%) | 0 (0%) | 0 (0%) | 1 (14.2%) | |
| Chemical industry | 17 (100%) | 0 (0%) | | 1 (5.8%) | 5 (29.4%) | 11 (64.7%) | | 10 (58.8%) | 2 (11.7%) | 1 (5.8%) | 4 (23.5%) | |
| Petroleum industry | 8 (80%) | 2 (20%) | | 3 (30%) | 0 (0%) | 7 (70%) | | 5 (50%) | 1 (10%) | 2 (20%) | 2 (20%) | |
| Services | 16 (38%) | 26 (61.9%) | <0.001 | 19 (45.2%) | 11 (26.1%) | 12 (28.5%) | 0.11 | 31 (73.8%) | 0 (0%) | 6 (14.2%) | 5 (11.9%) | 0.43 |

*(Continued)*

**Table 2.** (Continued)

| | | | | | | | | | | | | |
|---|---|---|---|---|---|---|---|---|---|---|---|---|
| Laundries | 6 (54.5%) | 5 (45.4%) | | 4 (36.3%) | 3 (27.2%) | 4 (36.3%) | | 8 (72.7%) | 0 (0%) | 2 (18.1%) | 1 (9%) | |
| Hairdressing | 2 (22.2%) | 7 (77.7%) | | 5 (55.5%) | 4 (44.4%) | 0 (0%) | | 8 (88.8%) | 0 (0%) | 1 (11.1%) | 0 (0%) | |
| Health care | 11 (39.2%) | 17 (60.7%) | | 13 (46.4%) | 6 (21.4%) | 9 (32.1%) | | 20 (71.4%) | 0 (0%) | 4 (14.2%) | 4 (14.2%) | |
| Farming, gardening | 25 (86.2%) | 4 (13.7%) | 0.25 | 12 (40%) | 7 (23.3%) | 11 (36.6%) | 0.49 | 20 (66.6%) | 0 (0%) | 8 (26.6%) | 2 (6.6%) | 0.43 |
| Agriculture | 18 (94.7%) | 1 (5.2%) | | 7 (36.8%) | 4 (21%) | 8 (42.1%) | | 12 (63.1%) | 0 (0%) | 6 (31.5%) | 1 (5.2%) | |
| Horticulture | 15 (83.3%) | 3 (16.6%) | | 9 (47.3%) | 6 (31.5%) | 4 (21%) | | 15 (78.9%) | 0 (0%) | 3 (15.7%) | 1 (5.2%) | |
| Building trade | 84 (97.6%) | 2 (2.3%) | <0.001 | 23 (26.4%) | 30 (34.4%) | 34 (39%) | 0.54 | 56 (65.1%) | 3 (3.4%) | 21 (24.4%) | 6 (6.9%) | 0.29 |
| Construction | 63 (98.4%) | 1 (1.5%) | | 16 (25%) | 21 (32.8%) | 27 (42.1%) | | 42 (66.6%) | 2 (3.1%) | 14 (22.2%) | 5 (7.9%) | |
| Painting | 36 (97.2%) | 1 (2.7%) | | 9 (23.6%) | 16 (42.1%) | 13 (34.2%) | | 23 (60.5%) | 1 (2.6%) | 12 (31.5%) | 2 (5.2%) | |
| Transport and related | 115 (93.4%) | 8 (6.5%) | <0.001 | 27 (21.9%) | 40 (32.5%) | 56 (45.5%) | 0.03 | 71 (57.7%) | 6 (4.8%) | 33 (26.8%) | 13 (10.5%) | 0.05 |
| Garages | 42 (95.4%) | 2 (4.5%) | | 6 (13.6%) | 18 (40.9%) | 20 (45.4%) | | 22 (50%) | 2 (4.5%) | 16 (36.3%) | 4 (9%) | |
| Nuclear power | 2 (100%) | 0 (0%) | | 0 (0%) | 0 (0%) | 2 (100%) | | 1 (50%) | 0 (0%) | 1 (50%) | 0 (0%) | |
| Driving jobs | 85 (94.4%) | 5 (5.5%) | | 18 (20%) | 31 (34.4%) | 41 (45.5%) | | 52 (57.7%) | 5 (5.5%) | 25 (27.7%) | 8 (8.8%) | |
| Warehousing | 29 (96.6%) | 1 (3.3%) | | 9 (30%) | 7 (23.3%) | 14 (46.6%) | | 16 (53.3%) | 0 (0%) | 10 (33.3%) | 4 (13.3%) | |
| Engine repairs | 36 (100%) | 0 (0%) | | 4 (11.1%) | 17 (47.2%) | 15 (41.6%) | | 19 (52.7%) | 2 (5.5%) | 13 (36.1%) | 2 (5.5%) | |

contact (all p<0.05). Tasks that included welding, mineral oil lubricants, the use of protective resins and diesel contact also had higher than expected staged cancers (all p<0.05). Conversely, higher stage cancers only were seen with the use of cement and the making of plastic foam.

## Clinical outcomes and occupational history

We compared the occupational histories with treatment outcomes and observed various interesting associations (Fig 1A–1D). The occupation that was performed for the longest period was the occupation that was used in the analysis when compared to clinical outcomes. For example, workers exposed to diesel fuels or fumes (Cox, HR 1.97 (95% CI 1.31–2.98) p = 0.001), or employed in a garage (HR 2.19 (95% CI 1.31–3.63) p = 0.001) were more likely to have disease progression and receive radical treatment (HR 1.75 (95% CI 1.23–2.47) p = 0.002) than others (Fig 1A and 1B). Participants undertaking plumbing/gas fitting/ventilation were also more likely to have disease progression (HR 2.15 (95% CI 1.15–4.01) p = 0.017) and receive radical treatment (HR 2.28 (95% CI 1.39–3.72) p = 0.003) than expected. Higher than expected progression (HR 2.36 (95% CI 1.19–469) p = 0.014) and radical treatment rates (HR 1.89 (95% CI 1.02–3.49) p = 0.04) were also seen in workers making/handling rubber products, whilst progression and radical treatment was more common in participants undertaking welding (HR 1.85 (95% CI 1.24–2.77) p = 0.003) and exposed to welding materials (HR 1.92 (95% CI 1.27–2.91) p = 0.002), than expected (Fig 1C). Consequently these workers (HR 1.85 (95% CI 1.24–2.77) p = 0.003), and those involved in smelting (HR 1.80 (95% CI 1.11–2.91) p = 0.016), were more likely to receive radical treatment than others. Higher than

**Table 3. Substance exposure compared to patient sex and tumour grade/stage.**

| | Gender | | | Grade | | | | Stage | | | | |
|---|---|---|---|---|---|---|---|---|---|---|---|---|
| | Male | Female | Chi sq P | 1 | 2 | 3 | Chi sq P | Ta | Tis | T1 | T2-4 | Chi sq P |
| Dyes | 9 (75%) | 3 (25%) | 0.83 | 2 (16.6%) | 7 (58.3%) | 3 (25%) | 0.11 | 10 (83.3%) | 0 (0%) | 2 (16.6%) | 0 (0%) | 0.46 |
| Crack-detection dyes | 29 (96.6%) | 1 (3.3%) | **0.01** | 4 (12.9%) | 16 (51.6%) | 11 (35.4%) | **0.02** | 23 (74.1%) | 2 (6.4%) | 5 (16.1%) | 1 (3.2%) | 0.28 |
| Dyeing material | 15 (93.7%) | 1 (6.2%) | 0.11 | 2 (12.5%) | 6 (37.5%) | 8 (50%) | 0.26 | 9 (56.2%) | 0 (0%) | 4 (25%) | 3 (18.7%) | 0.67 |
| Any other type of dye or stain | 18 (81.8%) | 4 (18.1%) | 0.62 | 5 (20.8%) | 8 (33.3%) | 11 (45.8%) | 0.53 | 13 (54.1%) | 0 (0%) | 5 (20.8%) | 6 (25%) | 0.2 |
| Cadmium | 16 (88.8%) | 2 (11.1%) | 0.24 | 2 (11.1%) | 7 (38.8%) | 9 (50%) | 0.18 | 8 (44.4%) | 1 (5.5%) | 7 (38.8%) | 2 (11.1%) | 0.16 |
| Chromium | 29 (90.6%) | 3 (9.3%) | 0.07 | 3 (9.3%) | 11 (34.3%) | 18 (56.2%) | **0.02** | 16 (50%) | 3 (9.3%) | 6 (18.7%) | 7 (21.8%) | **0.05** |
| Coal, gas and oil by product chemicals | 83 (100%) | 0 (0%) | **<0.001** | 16 (19.2%) | 30 (36.1%) | 37 (44.5%) | **0.04** | 47 (56.6%) | 2 (2.4%) | 20 (24%) | 14 (16.8%) | 0.25 |
| Gas works sludge | 12 (100%) | 0 (0%) | 0.06 | 1 (8.3%) | 6 (50%) | 5 (41.6%) | 0.17 | 7 (58.3%) | 1 (8.3%) | 1 (8.3%) | 3 (25%) | 0.33 |
| Coking plant fumes or residues | 48 (100%) | 0 (0%) | **<0.001** | 12 (25%) | 13 (27%) | 23 (47.9%) | 0.34 | 27 (57.4%) | 2 (4.2%) | 9 (19.1%) | 9 (19.1%) | 0.46 |
| Coal or coal products | 67 (98.5%) | 1 (1.4%) | **<0.001** | 18 (26.4%) | 17 (25%) | 33 (48.5%) | 0.17 | 34 (50%) | 3 (4.4%) | 18 (26.4%) | 13 (19.1%) | **0.05** |
| Cooking fumes | 23 (58.9%) | 16 (41%) | **<0.001** | 13 (32.5%) | 11 (27.5%) | 16 (40%) | 0.9 | 25 (62.5%) | 1 (2.5%) | 6 (15%) | 8 (20%) | 0.44 |
| Diesel exhaust fumes | 90 (96.7%) | 3 (3.2%) | **<0.001** | 21 (22.1%) | 27 (28.4%) | 47 (49.4%) | **0.03** | 49 (51.5%) | 7 (7.3%) | 24 (25.2%) | 15 (15.7%) | **0.01** |
| Oily/greasy rust proofing chemicals | 62 (95.3%) | 3 (4.6%) | **<0.001** | 13 (19.4%) | 21 (31.3%) | 33 (49.2%) | 0.05 | 39 (58.2%) | 2 (2.9%) | 16 (23.8%) | 10 (14.9%) | 0.62 |
| Diesel fuel | 79 (98.7%) | 1 (1.2%) | **<0.001** | 13 (16%) | 27 (33.3%) | 41 (50.6%) | **<0.001** | 35 (43.2%) | 6 (7.4%) | 27 (33.3%) | 13 (16%) | **<0.001** |
| Aircraft fuel | 9 (100%) | 0 (0%) | 0.1 | 1 (11.1%) | 1 (11.1%) | 7 (77.7%) | **0.05** | 5 (55.5%) | 1 (11.1%) | 2 (22.2%) | 1 (11.1%) | 0.6 |
| DDM or MDA | 1 (100%) | 0 (0%) | 0.6 | 0 (0%) | 1 (100%) | 0 (0%) | 0.3 | 1 (100%) | 0 (0%) | 0 (0%) | 0 (0%) | 0.9 |
| MOCA | 1 (100%) | 0 (0%) | 0.6 | 0 (0%) | 1 (100%) | 0 (0%) | 0.3 | 1 (100%) | 0 (0%) | 0 (0%) | 0 (0%) | 0.9 |
| Printers' ink | 25 (83.3%) | 5 (16.6%) | 0.43 | 10 (33.3%) | 11 (36.6%) | 9 (30%) | 0.61 | 20 (66.6%) | 0 (0%) | 5 (16.6%) | 5 (16.6%) | 0.64 |
| Solvents e.g. trichloroethylene | 112 (91.8%) | 10 (8.1%) | **<0.001** | 23 (18.4%) | 48 (38.4%) | 54 (43.2%) | **<0.001** | 79 (63.2%) | 2 (1.6%) | 26 (20.8%) | 18 (14.4%) | 0.5 |
| Arsenic | 9 (90%) | 1 (10%) | 0.34 | 2 (20%) | 6 (60%) | 2 (20%) | 0.13 | 7 (70%) | 1 (10%) | 1 (10%) | 1 (10%) | 0.58 |
| Fungicide, wood preservative (e.g. | 35 (100%) | 0 (0%) | **<0.001** | 7 (20%) | 9 (25.7%) | 19 (54.2%) | 0.11 | 18 (51.4%) | 1 (2.8%) | 11 (31.4%) | 5 (14.2%) | 0.27 |
| o-toluidine | 2 (100%) | 0 (0%) | 0.45 | 1 (50%) | 1 (50%) | 0 (0%) | 0.54 | 2 (100%) | 0 (0%) | 0 (0%) | 0 (0%) | 0.78 |
| 4-aminobiphenyl | 0 (0%) | 0 (0%) | NA | 0 (0%) | 0 (0%) | 0 (0%) | NA | 0 (0%) | 0 (0%) | 0 (0%) | 0 (0%) | NA |
| Ionising radiation (radioactive sources) | 11 (91.6%) | 1 (8.3%) | 0.23 | 3 (25%) | 2 (16.6%) | 7 (58.3%) | 0.33 | 6 (50%) | 0 (0%) | 2 (16.6%) | 4 (33.3%) | 0.15 |
| Coal tar cream | 14 (82.3%) | 3 (17.6%) | 0.63 | 8 (47%) | 2 (11.7%) | 7 (41.1%) | 0.17 | 11 (64.7%) | 1 (5.8%) | 2 (11.7%) | 3 (17.6%) | 0.73 |

Abbreviations: DDM–n-Dodecyl β-D-maltoside, MOCA–Methylene bis 2,4 aniline, MDA– 4,4'-methylenedianiline.

expected radical treatment rates were also seen in workers making/using cement (HR 1.85 (95% CI 1.24–2.73) p = 0.002). Finally, fewer than expected deaths were seen in healthcare workers (HR 0.17 (95% CI 0.04–0.70) p = 0.014) suggesting improved health (Fig 1D).

**Table 4. Occupational tasks compared to patient sex and tumour phenotype.**

| | Gender | | | Grade | | | | Stage | | | | |
|---|---|---|---|---|---|---|---|---|---|---|---|---|
| | Male | Female | Chi sq P | 1 | 2 | 3 | Chi sq P | Ta | Tis | T1 | T2-4 | Chi sq P |
| Smelting metals | 17 (100%) | 0 (0%) | **0.02** | 3 (17.6%) | 3 (17.6%) | 11 (64.7%) | 0.07 | 8 (47%) | 2 (11.7%) | 5 (29.4%) | 2 (11.7%) | 0.13 |
| Smelting metals nearby | 34 (100%) | 0 (0%) | **<0.001** | 7 (20%) | 8 (22.8%) | 20 (57.1%) | 0.06 | 16 (45.7%) | 2 (5.7%) | 9 (25.7%) | 8 (22.8%) | 0.07 |
| Assembling and repairing electrical goods | 30 (93.7%) | 2 (6.2%) | **0.02** | 8 (25%) | 9 (28.1%) | 15 (46.8%) | 0.56 | 19 (59.3%) | 2 (6.2%) | 4 (12.5%) | 7 (21.8%) | 0.21 |
| Assembling and repairing electrical goods nearby | 17 (100%) | 0 (0%) | **0.02** | 1 (5.8%) | 7 (41.1%) | 9 (52.9%) | **0.07** | 8 (47%) | 0 (0%) | 4 (23.5%) | 5 (29.4%) | 0.12 |
| Making products containing cadmium | 5 (100%) | 0 (0%) | 0.23 | 2 (33.3%) | 0 (0%) | 4 (66.6%) | 0.21 | 3 (50%) | 1 (16.6%) | 2 (33.3%) | 0 (0%) | 0.18 |
| Making products containing cadmium nearby | 7 (100%) | 0 (0%) | 0.15 | 1 (14.2%) | 3 (42.8%) | 3 (42.8%) | 0.60 | 5 (71.4%) | 0 (0%) | 2 (28.5%) | 0 (0%) | 0.69 |
| Making or using cement | 63 (100%) | 0 (0%) | **<0.001** | 14 (22.2%) | 19 (30.1%) | 30 (47.6%) | 0.17 | 31 (50%) | 6 (9.6%) | 16 (25.8%) | 9 (14.5%) | **<0.001** |
| Making or using cement nearby | 29 (93.5%) | 2 (6.4%) | **0.03** | 6 (19.3%) | 9 (29%) | 16 (51.6%) | 0.22 | 16 (55.1%) | 1 (3.4%) | 8 (27.5%) | 4 (13.7%) | 0.67 |
| Soldering or brazing | 56 (96.5%) | 2 (3.4%) | **<0.001** | 12 (20.3%) | 18 (30.5%) | 29 (49.1%) | 0.10 | 31 (52.5%) | 2 (3.3%) | 15 (25.4%) | 11 (18.6%) | 0.17 |
| Soldering or brazing nearby | 31 (96.8%) | 1 (3.1%) | **0.01** | 9 (26.4%) | 12 (35.2%) | 13 (38.2%) | 0.78 | 22 (64.7%) | 2 (5.8%) | 5 (14.7%) | 5 (14.7%) | 0.71 |
| Metal plating | 11 (84.6%) | 2 (15.3%) | 0.54 | 3 (23%) | 3 (23%) | 7 (53.8%) | 0.50 | 6 (46.1%) | 1 (7.6%) | 4 (30.7%) | 2 (15.3%) | 0.48 |
| Metal plating nearby | 10 (100%) | 0 (0%) | 0.09 | 3 (30%) | 3 (30%) | 4 (40%) | 0.99 | 7 (70%) | 1 (10%) | 1 (10%) | 1 (10%) | 0.58 |
| Cadmium plating | 2 (100%) | 0 (0%) | 0.45 | 0 (0%) | 2 (100%) | 0 (0%) | 0.11 | 2 (100%) | 0 (0%) | 0 (0%) | 0 (0%) | 0.78 |
| Cadmium plating nearby | 6 (100%) | 0 (0%) | 0.18 | 1 (16.6%) | 1 (16.6%) | 4 (66.6%) | 0.35 | 4 (66.6%) | 1 (16.6%) | 1 (16.6%) | 0 (0%) | 0.25 |
| Fumes from quenching (heat treatment) | 33 (97%) | 1 (2.9%) | **0.01** | 13 (36.1%) | 6 (16.6%) | 17 (47.2%) | 0.16 | 22 (61.1%) | 3 (8.3%) | 8 (22.2%) | 3 (8.3%) | 0.29 |
| Fumes from quenching (heat treatment) nearby | 54 (94.7%) | 3 (5.2%) | **<0.001** | 12 (20.6%) | 18 (31%) | 28 (48.2%) | 0.13 | 31 (53.4%) | 3 (5.1%) | 14 (24.1%) | 10 (17.2%) | 0.25 |
| Fumes from forging | 32 (100%) | 0 (0%) | **<0.001** | 9 (26.4%) | 12 (35.2%) | 13 (38.2%) | 0.78 | 23 (67.6%) | 1 (2.9%) | 6 (17.6%) | 4 (11.7%) | 0.99 |
| Fumes from forging nearby | 43 (97.7%) | 1 (2.2%) | **<0.001** | 9 (19.5%) | 14 (30.4%) | 23 (50%) | 0.13 | 24 (53.3%) | 2 (4.4%) | 10 (22.2%) | 9 (20%) | 0.28 |
| Crack detection /Non-destructive testing | 23 (100%) | 0 (0%) | **0.01** | 3 (12.5%) | 10 (41.6%) | 11 (45.8%) | 0.13 | 15 (62.5%) | 1 (4.1%) | 5 (20.8%) | 3 (12.5%) | 0.99 |
| Crack detection /Non-destructive testing nearby | 19 (95%) | 1 (5%) | 0.06 | 2 (10%) | 8 (40%) | 10 (50%) | 0.12 | 11 (55%) | 2 (10%) | 3 (15%) | 4 (20%) | 0.22 |
| Resins in 'cold box' techniques in foundries | 4 (100%) | 0 (0%) | 0.28 | 0 (0%) | 2 (50%) | 2 (50%) | 0.39 | 2 (50%) | 1 (25%) | 1 (25%) | 0 (0%) | **0.09** |
| Resins in 'cold box' techniques in foundries nearby | 4 (100%) | 0 (0%) | 0.28 | 0 (0%) | 2 (50%) | 2 (50%) | 0.39 | 2 (50%) | 0 (0%) | 1 (25%) | 1 (25%) | 0.83 |
| Contact with weld material and steel | 65 (98.4%) | 1 (1.5%) | **<0.001** | 11 (16.1%) | 20 (29.4%) | 37 (54.4%) | **<0.001** | 37 (54.4%) | 5 (7.3%) | 19 (27.9%) | 7 (10.2%) | **0.04** |
| Contact with weld material and steel nearby | 45 (95.7%) | 2 (4.2%) | **<0.001** | 7 (14.8%) | 19 (40.4%) | 21 (44.6%) | **0.04** | 24 (52.1%) | 3 (6.5%) | 9 (19.5%) | 10 (21.7%) | 0.09 |
| Fume from producing and using coke, and converting coal to gas. | 20 (100%) | 0 (0%) | **0.01** | 3 (15%) | 5 (25%) | 12 (60%) | 0.10 | 10 (50%) | 2 (10%) | 4 (20%) | 4 (20%) | 0.20 |
| Fume from producing and using coke, and converting coal to gas. nearby | 24 (100%) | 0 (0%) | **0.01** | 3 (12.5%) | 10 (41.6%) | 11 (45.8%) | 0.13 | 11 (45.8%) | 1 (4.1%) | 8 (33.3%) | 4 (16.6%) | 0.23 |

*(Continued)*

**Table 4.** (*Continued*)

| | Gender | | | Grade | | | | Stage | | | | |
|---|---|---|---|---|---|---|---|---|---|---|---|---|
| | Male | Female | Chi sq P | 1 | 2 | 3 | Chi sq P | Ta | Tis | T1 | T2-4 | Chi sq P |
| Residues from coke and gas production | 23 (95.8%) | 1 (4.1%) | **0.03** | 4 (16.6%) | 7 (29.1%) | 13 (54.1%) | 0.18 | 12 (50%) | 2 (8.3%) | 7 (29.1%) | 3 (12.5%) | 0.26 |
| Residues from coke and gas production nearby | 18 (100%) | 0 (0%) | **0.02** | 5 (27.7%) | 4 (22.2%) | 9 (50%) | 0.55 | 9 (50%) | 1 (5.5%) | 6 (33.3%) | 2 (11.1%) | 0.43 |
| Making or handling plastics | 23 (92%) | 2 (8%) | 0.08 | 9 (34.6%) | 9 (34.6%) | 8 (30.7%) | 0.72 | 16 (61.5%) | 2 (7.6%) | 4 (15.3%) | 4 (15.3%) | 0.55 |
| Making or handling plastics nearby | 13 (92.8%) | 1 (7.1%) | 0.16 | 3 (21.4%) | 4 (28.5%) | 7 (50%) | 0.61 | 9 (64.2%) | 1 (7.1%) | 1 (7.1%) | 3 (21.4%) | 0.43 |
| Making or handling rubber products | 20 (100%) | 0 (0%) | **0.01** | 3 (14.2%) | 8 (38%) | 10 (47.6%) | 0.24 | 12 (57.1%) | 1 (4.7%) | 4 (19%) | 4 (19%) | 0.76 |
| Making or handling rubber products nearby | 8 (100%) | 0 (0%) | 0.13 | 2 (25%) | 3 (37.5%) | 3 (37.5%) | 0.90 | 6 (75%) | 0 (0%) | 0 (0%) | 2 (25%) | 0.38 |
| Breakdown of resins used to make moulds and cores | 15 (100%) | 0 (0%) | **0.03** | 2 (13.3%) | 5 (33.3%) | 8 (53.3%) | 0.28 | 6 (40%) | 1 (6.6%) | 5 (33.3%) | 3 (20%) | 0.23 |
| Breakdown of resins used to make moulds and cores nearby | 4 (80%) | 1 (20%) | 0.89 | 1 (20%) | 2 (40%) | 2 (40%) | 0.84 | 4 (80%) | 0 (0%) | 1 (20%) | 0 (0%) | 0.81 |
| Making chemicals from coal, coke, oil and gas byproducts | 17 (100%) | 0 (0%) | **0.02** | 4 (23.5%) | 3 (17.6%) | 10 (58.8%) | 0.20 | 9 (56.2%) | 2 (12.5%) | 3 (18.7%) | 2 (12.5%) | 0.22 |
| Making chemicals from coal, coke, oil and gas byproducts nearby | 14 (93.3%) | 1 (6.6%) | 0.14 | 3 (20%) | 6 (40%) | 6 (40%) | 0.59 | 10 (66.6%) | 0 (0%) | 4 (26.6%) | 1 (6.6%) | 0.72 |
| e.g. additives to aeroplane fuel | 3 (100%) | 0 (0%) | 0.35 | 2 (66.6%) | 0 (0%) | 1 (33.3%) | 0.34 | 3 (100%) | 0 (0%) | 0 (0%) | 0 (0%) | 0.65 |
| e.g. additives to aeroplane fuel nearby | 2 (100%) | 0 (0%) | 0.45 | 1 (50%) | 1 (50%) | 0 (0%) | 0.54 | 2 (100%) | 0 (0%) | 0 (0%) | 0 (0%) | 0.78 |
| Mineral oils used as lubricants and coolants | 61 (98.3%) | 1 (1.6%) | **<0.001** | 9 (14.2%) | 24 (38%) | 30 (47.6%) | **0.01** | 30 (47.6%) | 6 (9.5%) | 18 (28.5%) | 9 (14.2%) | **0.00** |
| Mineral oils used as lubricants and coolants nearby | 32 (94.1%) | 2 (5.8%) | **0.02** | 7 (20.5%) | 12 (35.2%) | 15 (44.1%) | 0.39 | 21 (63.6%) | 0 (0%) | 6 (18.1%) | 6 (18.1%) | 0.53 |
| Making and using resins | 28 (100%) | 0 (0%) | **<0.001** | 3 (10.3%) | 13 (44.8%) | 13 (44.8%) | **0.04** | 16 (55.1%) | 0 (0%) | 12 (41.3%) | 1 (3.4%) | **0.01** |
| Making and using resins nearby | 12 (92.3%) | 1 (7.6%) | 0.20 | 2 (15.3%) | 3 (23%) | 8 (61.5%) | 0.20 | 7 (53.8%) | 2 (15.3%) | 4 (30.7%) | 0 (0%) | **0.03** |
| Making plastic foam | 2 (66.6%) | 1 (33.3%) | 0.65 | 2 (66.6%) | 0 (0%) | 1 (33.3%) | 0.34 | 3 (100%) | 0 (0%) | 0 (0%) | 0 (0%) | 0.65 |
| Making plastic foam nearby | 6 (100%) | 0 (0%) | 0.18 | 1 (16.6%) | 3 (50%) | 2 (33.3%) | 0.56 | 3 (50%) | 2 (33.3%) | 1 (16.6%) | 0 (0%) | **0.00** |
| Degreasing | 55 (98.2%) | 1 (1.7%) | **<0.001** | 12 (20.6%) | 20 (34.4%) | 26 (44.8%) | 0.19 | 32 (55.1%) | 2 (3.4%) | 15 (25.8%) | 9 (15.5%) | 0.41 |
| Degreasing nearby | 30 (93.7%) | 2 (6.2%) | **0.02** | 7 (21.8%) | 11 (34.3%) | 14 (43.7%) | 0.51 | 20 (62.5%) | 1 (3.1%) | 5 (15.6%) | 6 (18.7%) | 0.69 |
| Dry-cleaning | 4 (66.6%) | 2 (33.3%) | 0.52 | 0 (0%) | 3 (50%) | 3 (50%) | 0.24 | 3 (50%) | 0 (0%) | 2 (33.3%) | 1 (16.6%) | 0.78 |
| Dry-cleaning nearby | 2 (66.6%) | 1 (33.3%) | 0.65 | 0 (0%) | 0 (0%) | 3 (100%) | 0.09 | 2 (66.6%) | 0 (0%) | 0 (0%) | 1 (33.3%) | 0.62 |
| Timber treatment | 21 (95.4%) | 1 (4.5%) | **0.04** | 6 (27.2%) | 4 (18.1%) | 12 (54.5%) | 0.23 | 13 (59%) | 1 (4.5%) | 6 (27.2%) | 2 (9%) | 0.77 |
| Timber treatment nearby | 9 (100%) | 0 (0%) | 0.10 | 2 (22.2%) | 3 (33.3%) | 4 (44.4%) | 0.84 | 5 (55.5%) | 1 (11.1%) | 2 (22.2%) | 1 (11.1%) | 0.60 |
| Plumbing, gas-fitting, heat and ventilation fitting | 29 (100%) | 0 (0%) | **<0.001** | 5 (17.2%) | 7 (24.1%) | 17 (58.6%) | 0.06 | 13 (46.4%) | 3 (10.7%) | 6 (21.4%) | 6 (21.4%) | **0.03** |
| Plumbing, gas-fitting, heat and ventilation fitting nearby | 13 (100%) | 0 (0%) | **0.05** | 4 (30.7%) | 3 (23%) | 6 (46.1%) | 0.79 | 7 (58.3%) | 0 (0%) | 2 (16.6%) | 3 (25%) | 0.54 |

(*Continued*)

**Table 4.** (Continued)

| | Gender | | | Grade | | | | Stage | | | | |
|---|---|---|---|---|---|---|---|---|---|---|---|---|
| | Male | Female | Chi sq P | 1 | 2 | 3 | Chi sq P | Ta | Tis | T1 | T2-4 | Chi sq P |
| Painting | 31 (88.5%) | 4 (11.4%) | 0.10 | 9 (25.7%) | 15 (42.8%) | 11 (31.4%) | 0.27 | 21 (61.7%) | 1 (2.9%) | 9 (26.4%) | 3 (8.8%) | 0.72 |
| Painting nearby | 18 (90%) | 2 (10%) | 0.17 | 4 (20%) | 8 (40%) | 8 (40%) | 0.49 | 12 (63.1%) | 1 (5.2%) | 4 (21%) | 2 (10.5%) | 0.96 |
| Contact with industrial diesel | 35 (100%) | 0 (0%) | <**0.001** | 4 (11.1%) | 11 (30.5%) | 21 (58.3%) | **0.01** | 13 (36.1%) | 4 (11.1%) | 14 (38.8%) | 5 (13.8%) | <**0.001** |
| Contact with industrial diesel nearby | 11 (100%) | 0 (0%) | 0.07 | 2 (18.1%) | 3 (27.2%) | 6 (54.5%) | 0.49 | 7 (63.6%) | 0 (0%) | 1 (9%) | 3 (27.2%) | 0.38 |
| Separated out impurities, ores, scrap or wastes | 11 (91.6%) | 1 (8.3%) | 0.23 | 3 (25%) | 4 (33.3%) | 5 (41.6%) | 0.90 | 5 (41.6%) | 1 (8.3%) | 5 (41.6%) | 1 (8.3%) | 0.16 |
| Separated out impurities, ores, scrap or wastes nearby | 4 (100%) | 0 (0%) | 0.28 | 1 (25%) | 1 (25%) | 2 (50%) | 0.89 | 1 (25%) | 0 (0%) | 2 (50%) | 1 (25%) | 0.31 |
| Pesticide and herbicide treatments | 10 (100%) | 0 (0%) | 0.09 | 2 (20%) | 3 (30%) | 5 (50%) | 0.68 | 7 (70%) | 0 (0%) | 2 (20%) | 1 (10%) | 0.94 |
| Pesticide and herbicide treatments nearby | 5 (100%) | 0 (0%) | 0.23 | 0 (0%) | 2 (40%) | 3 (60%) | 0.31 | 1 (20%) | 0 (0%) | 3 (60%) | 1 (20%) | 0.10 |
| Burning plastics | 10 (100%) | 0 (0%) | 0.09 | 3 (30%) | 2 (20%) | 5 (50%) | 0.68 | 6 (60%) | 0 (0%) | 2 (20%) | 2 (20%) | 0.83 |
| Burning plastics nearby | 6 (100%) | 0 (0%) | 0.18 | 2 (33.3%) | 2 (33.3%) | 2 (33.3%) | 0.97 | 4 (66.6%) | 0 (0%) | 1 (16.6%) | 1 (16.6%) | 0.96 |
| Radiotherapy | 4 (66.6%) | 2 (33.3%) | 0.52 | 2 (33.3%) | 1 (16.6%) | 3 (50%) | 0.73 | 3 (50%) | 0 (0%) | 2 (33.3%) | 1 (16.6%) | 0.78 |
| Radiotherapy nearby | 2 (100%) | 0 (0%) | 0.45 | 1 (50%) | 1 (50%) | 0 (0%) | 0.54 | 2 (100%) | 0 (0%) | 0 (0%) | 0 (0%) | 0.78 |
| Industrial radiography | 3 (100%) | 0 (0%) | 0.35 | 0 (0%) | 1 (33.3%) | 2 (66.6%) | 0.45 | 1 (33.3%) | 0 (0%) | 1 (33.3%) | 1 (33.3%) | 0.58 |
| Industrial radiography nearby | 4 (80%) | 1 (20%) | 0.89 | 2 (40%) | 1 (20%) | 2 (40%) | 0.85 | 5 (100%) | 0 (0%) | 0 (0%) | 0 (0%) | 0.44 |

## Discussion

We report the outcomes from BC in consecutive patient cohort recruited in South Yorkshire, UK. We find a variety of workers with BC with high risks for aggressive disease that need radical treatment. There are several key findings that require discussion. Firstly, the occupational classes, tasks and contacts reflect local industrial patterns. Most men were employed in the steel, engineering, mining and building sectors, and few worked in industries more typical for BC (with aromatic amine contact); such as rubber, printing, painting and textile sectors. The carcinogens within our population are likely to be a mixture of PAHs, diesel fumes and combustion products. We did find some aromatic amines in occult use in the engineering and metal industries (such as crack detection dyes for non-destructive testing [13]), but these appeared uncommon. PAH exposure arises through cutaneous contact with lubricants, oils and metal working fluids, or inhalation of fumes or combustion products. Our findings contrast and complement a recent systematic review of occupational BC within the UK we conducted [8]. Within this review of 703,941 persons, we found the highest incidence of BC was in chemical process, rubber and dye workers, whilst electrical, transport and chemical process workers had the highest risks of death from BC. Our current data show that electrical workers have a high risk of developing aggressive BC and focus this risk on tasks such as welding and soldering. Fumes from these tasks contain lead oxide, heavy metals (arsenic, cadmium, chromium and nickel etc.) and colophony (rosin based flux containing acetone and carbon

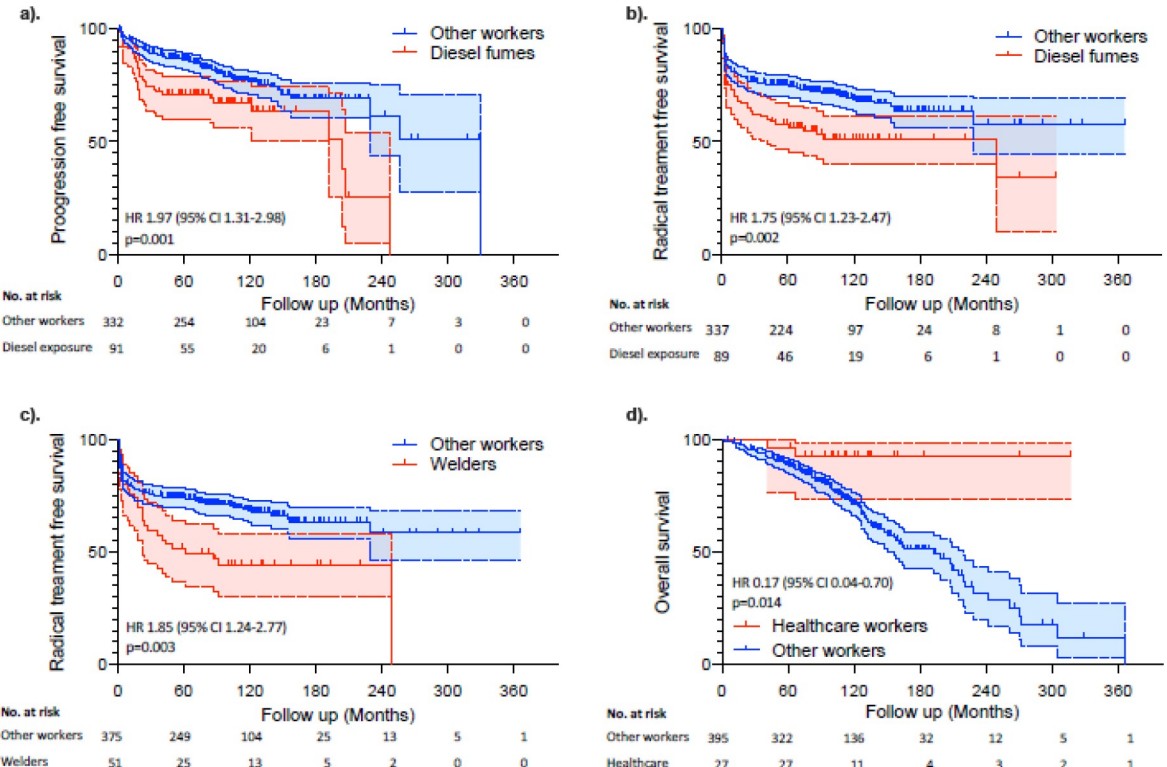

**Fig 1.** a. Progression free survival of bladder cancer of patients exposed and not exposed to occupational diesel fumes. b. Radical treatment free survival of bladder cancer of patients exposed and not exposed to occupational diesel fumes. c. Radical treatment free survival of bladder cancer of patients exposed and not exposed to occupational welding. d. Overall survival of bladder cancer from patients who were healthcare workers compared to any other form of work.

monoxide). Our observations may partly explain the high prevalence and mortality from BC seen in Yorkshire [8].

Secondly, our data support the carcinogenicity of diesel fumes to the urothelium. Previous reports have examined this systematically [23] and in 2012 the IARC classified diesel exhaust fumes as carcinogenic (class 1) to the lung (with '*sufficient evidence*') and the bladder (with '*limited evidence*') [24]. We add to these data by showing that contact with diesel fuels and fumes were associated with high grade/high stage BC and higher risks of disease progression. Reflecting employment patterns, diesel contact was more common in men than women, and there was some evidence of a dose interaction with cigarette smoking (workers with diesel exhaust exposure had higher pack years (mean: 41 ± 34) than those without (mean 30 ± 22.9, T test p = 0.008)). Workers with diesel exposure were commonly employed in the welding, soldering, agriculture, building, transport and engine repair sectors, and undertook typical task for these sectors (e.g. driving, mixing cement, welding). It is also worth noting that diesel exposure and garage work can also occur with hobbies, reflecting an additional exposure.

Thirdly, our data suggest that occupational history should be included in the BC care pathway. BC is one of the commonest human cancers and one of the most expensive to manage. Much of this expense is spent on monitoring patients with NMI cancers or in screening people with non-visible haematuria [2,25]. Better targeting of resource, with improved survival, more effective screening and lower costs, could be achieved if patient risk stratification was available [26]. Whilst current guidelines rely on age and extent of haematuria [3], our findings suggest that occupational history could guide clinicians to persons at risk of aggressive BC. For

example, screening of a few very-high risk persons, e.g. those with aristolochic acid exposure [27] or employees working with aromatic amines [28] is performed, but our data suggest occupational urothelial carcinogenic exposures are common and could help triage a population (by focusing upon the risks of aggressive BCs).

Fourthly, there were differences in exposures between men and women. These included distinct patterns of employment, differences in smoking rates and patterns (direct and passive ETS), hair dye use and hobbies. Given that most participants were male; our reported findings mostly reflect risk in men. Analysis of females only, suggests associations between high grade/ high stage BC and workers undertaking electroplating and cutlery manufacture, and tasks such as degreasing and painting ($p < 0.05$).

There are limitations to our work. Most importantly, the sample size was small and so this data should be viewed as hypothesis-generating, rather than definitive. Our aim was to undertake an explorative cohort study (rather than a clinical trial) and so no formal power calculation was performed. This reflects that very little is known about occupational risks and bladder cancer phenotypes and so powering was not possible. Our findings require validation in larger cohorts enriched for engineering and metal workers. Follow up was immature (median 8.4 years) in our series, and so many progressive tumours had not led to death in the participants. As such, we used stage and grade, progression and radical treatment, as surrogate measures for BC specific mortality. With longer follow up, we would look to see if these occupational tasks were associated with mortality or whether aggressive treatment could prevent this. Finally, the questionnaires were self-completed. Workers may have missed key exposures and others appeared more prominent that their actual workload. We asked participants to estimate the duration of each task, but these dates were often missing or very broad.

## Conclusions

We identified multiple occupational tasks and contacts associated with high grade and high stage BC. Workers exposed to diesel fumes, employed in a garage, undertaking plumbing/gas fitting/ventilation, welding were more likely to have disease progression and receive radical treatment than others. These findings require validation and could be used to risk stratify persons with haematuria or follow up of non-invasive BC.

## Supporting information

**S1 Fig. Survival Curves for PLOS One.**
(PDF)

**S1 File. No logo Sheff Occup Questionnaire v4 08 11 11.s**
(PDF)

## Author Contributions

**Conceptualization:** Jon Griffin, Aidan P. Noon, Syed Hussain, Marcus G. Cumberbatch, James W.F. Catto.

**Data curation:** Ibrahim Jubber, Jon Griffin, Aidan P. Noon, Syed Hussain, Marcus G. Cumberbatch, James W.F. Catto.

**Formal analysis:** Ibrahim Jubber, Marcus G. Cumberbatch, James W.F. Catto.

**Funding acquisition:** James W.F. Catto.

**Investigation:** Oliver Reed, Ibrahim Jubber, Jon Griffin, Aidan P. Noon, Louise Goodwin, Marcus G. Cumberbatch, James W.F. Catto.

**Methodology:** Oliver Reed, Ibrahim Jubber, Louise Goodwin, Marcus G. Cumberbatch, James W.F. Catto.

**Project administration:** Louise Goodwin, James W.F. Catto.

**Resources:** Louise Goodwin, James W.F. Catto.

**Software:** Ibrahim Jubber, Marcus G. Cumberbatch.

**Supervision:** Ibrahim Jubber, Marcus G. Cumberbatch, James W.F. Catto.

**Validation:** James W.F. Catto.

**Visualization:** James W.F. Catto.

**Writing – original draft:** Oliver Reed.

**Writing – review & editing:** James W.F. Catto.

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
