## [Decision Letter · Decision Letter 0]

7 May 2020

PONE-D-20-10549

Occupational Bladder cancer: A cross section survey of previous employments, tasks and exposures matched to cancer phenotypes.

PLOS ONE

Dear Dr Reed

Thank you for submitting your manuscript to PLOS ONE. After careful consideration, we feel that it has merit but does not fully meet PLOS ONE’s publication criteria as it currently stands. Therefore, we invite you to submit a revised version of the manuscript that addresses the points raised during the review process.

We would appreciate receiving your revised manuscript by 12th of June. To enhance the reproducibility of your results, we recommend that if applicable you deposit your laboratory protocols in protocols.io, where a protocol can be assigned its own identifier (DOI) such that it can be cited independently in the future. For instructions see: http://journals.plos.org/plosone/s/submission-guidelines#loc-laboratory-protocols

We look forward to receiving your revised manuscript.

Kind regards,

Eva Compérat

Academic Editor

PLOS ONE

Journal Requirements:

'All patients underwent informed consent in an ethically approved programme (South Yorkshire Research Ethics Committee approval number 10/H1310/73) before entry.'

a. Please amend your current ethics statement to confirm that your named institutional review board or ethics committee specifically approved this study.

3. Please include additional information regarding the survey or questionnaire used in the study and ensure that you have provided sufficient details that others could replicate the analyses.

For instance, if you developed a questionnaire as part of this study and it is not under a copyright more restrictive than CC-BY, please include a copy, in both the original language and English, as Supporting Information.

4. Please provide a sample size and power calculation in the Methods, or discuss the reasons for not performing one before study initiation.

5. Your ethics statement must appear in the Methods section of your manuscript. If your ethics statement is written in any section besides the Methods, please move it to the Methods section and delete it from any other section. Please also ensure that your ethics statement is included in your manuscript, as the ethics section of your online submission will not be published alongside your manuscript.

6. Please include a caption for figure 1.

7. Please include captions for your Supporting Information files at the end of your manuscript, and update any in-text citations to match accordingly. Please see our Supporting Information guidelines for more information: http://journals.plos.org/plosone/s/supporting-information

Reviewers' comments:

Reviewer's Responses to Questions

**Comments to the Author**

1. Is the manuscript technically sound, and do the data support the conclusions?

Reviewer #1: Yes

Reviewer #2: Yes

2. Has the statistical analysis been performed appropriately and rigorously? 

Reviewer #1: Yes

Reviewer #2: Yes

3. Have the authors made all data underlying the findings in their manuscript fully available?

Reviewer #1: Yes

Reviewer #2: Yes

4. Is the manuscript presented in an intelligible fashion and written in standard English?

Reviewer #1: Yes

Reviewer #2: Yes

5. Review Comments to the Author

Reviewer #1: The submitted manuscript “Occupational Bladder cancer: A cross section survey of previous employments, tasks and exposures matched to cancer phenotypes.” by Reed et al. reports on 454 patients with urothelial bladder cancer, first diagnosed and treated between 02/2010 - 07/2012 at a single institution, the RHH (Sheffield).

Based on a patient-reported questionnaire, patients were evaluated for potential carcinogen exposure, whilst occupational classes were assigned using NYK and ISCO-1958 codes.

With a median follow-up of 8.4 years, the authors additionally report on tumor progression and the need of radical intervention. Outcome data was collected between 08/2018 - 10/2018.

This questionnaire-based evaluation, revealed multiple occupational tasks and contacts associated with high grade and high stage BC. Tumors were classified according to TNM and WHO 1973 criteria, therefore G1-G3 grading data has been reported.

Typical for an urothelial cancer population is the distribution between men and women with a ratio of roughly 4:1. Therefore, the reported data refers to mainly men, since all patients were included at time of initial diagnosis of bladder cancer.

The authors found differences in exposures between men and women, including

distinct patterns of employment, differences in smoking rates and patterns,

hair dye use and hobbies.

When only female patients were analyzed, the data suggests an association between high grade/high stage BC and workers undertaking electroplating and cutlery manufacture, and tasks such as degreasing and painting.

Although, the included patient cohort quite small to evaluate potential influences of exposure to occupational carcinogens, there are some interesting findings, worth to be reported.

Limitations of the manuscript are well described (e.g. estimated duration of each task), data reported and analyzed with appropriate methods, and outcome data revealed quite distinct differences for specific occupational groups.

Overall, an interesting and well-written manuscript, worth publication after minor revision:

Some comments:

- Please use BOLD for all significant values, which makes it easier to read tables.

- For Kaplan-Meier curves, please add “at risk numbers” on the bottom of the graphs.

Reviewer #2: Paper Occupational bladder cancer: A cross section survey of previous employments, tasks and exposures matched to cancer phenotypes by dr. Reed et al.

Very interesting study. I have just few comments, which, as I believe, can improve paper.

1, I would stress time of exposure. I believe it is very important point missed in discussion and even in abstract

2, I would suggest also hobby of participants. It is well known from other epidemiological studies......example can be psittacosis. It is indeed mostly occupational exposure linked disease, however substantial part of patients came from hobby sector (bird breeders, parrot lovers, etc, etc). Authors listed garage work as potential risk factor for aggressive disease course with recquired radical treatment. There are many car lovers, bikers who spend a substantial time in care of their gear and indeed they are haevily exposed to risk substanties. I think this should be at least discussed.

3, Is there any chance to check any link between variant histology (mostly highly aggressive tumors, sometimes with bland cytology-ie LG, like nested UC) and occupational exposure

Thank you

6. PLOS authors have the option to publish the peer review history of their article (what does this mean?). If published, this will include your full peer review and any attached files.

Reviewer #1: No

Reviewer #2: No

---

## [Author Response · Author response to Decision Letter 0]

16 Jun 2020

3, Is there any chance to check any link between variant histology (mostly highly aggressive tumors, sometimes with bland cytology-ie LG, like nested UC) and occupational exposure

Answer: This is a very valuable point. Indeed some cases were defined as variants in their pathological reports. However, in 2010, the presence or absence of variant histology was not reliably reported in our hospital or in the UK (please see Urol Oncol. 2013 Nov;31(8):1650-5). As such, we are unable to reliably know whether each case had/did not have variant patterns and so have not reported this.

---

## [Decision Letter · Decision Letter 1]

4 Sep 2020

Occupational Bladder cancer: A cross section survey of previous employments, tasks and exposures matched to cancer phenotypes.

PONE-D-20-10549R1

Dear Dr. Reed,

We’re pleased to inform you that your manuscript has been judged scientifically suitable for publication and will be formally accepted for publication once it meets all outstanding technical requirements.

Kind regards,

Amitava Mukherjee, ME, Ph.D.

Academic Editor

PLOS ONE

Additional Editor Comments (optional):

Reviewers' comments:

Reviewer's Responses to Questions

**Comments to the Author**

1. If the authors have adequately addressed your comments raised in a previous round of review and you feel that this manuscript is now acceptable for publication, you may indicate that here to bypass the “Comments to the Author” section, enter your conflict of interest statement in the “Confidential to Editor” section, and submit your "Accept" recommendation.

Reviewer #1: All comments have been addressed

Reviewer #2: All comments have been addressed

2. Is the manuscript technically sound, and do the data support the conclusions?

Reviewer #1: Yes

Reviewer #2: Yes

3. Has the statistical analysis been performed appropriately and rigorously? 

Reviewer #1: Yes

Reviewer #2: Yes

4. Have the authors made all data underlying the findings in their manuscript fully available?

Reviewer #1: Yes

Reviewer #2: Yes

5. Is the manuscript presented in an intelligible fashion and written in standard English?

Reviewer #1: Yes

Reviewer #2: Yes

6. Review Comments to the Author

Reviewer #1: The authors have addressed the comments made by the reviewers and changed their manuscript accrodingly.

Reviewer #2: I believe this paper can help to solve questions about potential agents playing etiological role in development of UC. I do not have any further questions. Thank you

7. PLOS authors have the option to publish the peer review history of their article (what does this mean?). If published, this will include your full peer review and any attached files.

Reviewer #1: No

Reviewer #2: No

---

## [Editor Report · Acceptance letter]

30 Sep 2020

PONE-D-20-10549R1 

Occupational Bladder cancer: A cross section survey of previous employments, tasks and exposures matched to cancer phenotypes 

Dear Dr. Reed:

I'm pleased to inform you that your manuscript has been deemed suitable for publication in PLOS ONE. Congratulations! Your manuscript is now with our production department. 

Kind regards, 

on behalf of

Professor Dr. Amitava Mukherjee 

Academic Editor

PLOS ONE